# Molecular ferroelectric with low-magnetic-field magnetoelectricity at room temperature

Zhao-Bo Hu [1,2], Xinyu Yang[3], Jinlei Zhang [4] ✉, Ling-Ao Gui[2], Yi-Fan Zhang[2], Xiao-Dong Liu [1], Zi-Han Zhou[1], Yucheng Jiang [4], Yi Zhang [5] ✉, Shuai Dong [3] ✉ & You Song [1] ✉

Magnetoelectric materials, which encompass coupled magnetic and electric polarizabilities within a single phase, hold great promises for magnetic controlled electronic components or electric-field controlled spintronics. However, the realization of ideal magnetoelectric materials remains tough due to the inborn competion between ferroelectricity and magnetism in both levels of symmetry and electronic structure. Herein, we introduce a methodology for constructing single phase paramagnetic ferroelectric molecule [TMCM] [FeCl$_4$], which shows low-magnetic-field magnetoelectricity at room temperature. By applying a low magnetic field (≤1 kOe), the halogen Cl···Cl distance and the volume of [FeCl$_4$]$^-$ anions could be manipulated. This structural change causes a characteristic magnetostriction hysteresis, resulting in a substantial deformation of ~$10^{-4}$ along the $a$-axis under an in-plane magnetic field of 2 kOe. The magnetostrictive effect is further qualitatively simulated by density functional theory calculations. Furthermore, this mechanical deformation significantly dampens the ferroelectric polarization by directly influencing the overall dipole configuration. As a result, it induces a remarkable $\alpha_{31}$ component (~89 mV Oe$^{-1}$ cm$^{-1}$) of the magnetoelectric tensor. And the magnetoelectric coupling, characterized by the change of polarization, reaches ~12% under 40 kOe magnetic field. Our results exemplify a design methodology that enables the creation of room-temperature magnetoelectrics by leveraging the potent effects of magnetostriction.

Magnetoelectric (ME) coupling effect in materials offers a promising pathway for the advancement of high-density data storage, spintronics, and low-consumption nanoelectronics[1–6]. To enable feasible applications, it is crucial to control this coupling effect under a low magnetic field and at room temperature[7–10]. Until now, only a few materials have demonstrated unequivocal coupled magnetic and electric polarizabilities at room temperature[11]. Examples include perovskite-type bismuth ferrite (BiFeO$_3$)[12] and hexagonal $R$MnO$_3$[13]

($R$ represents rare earth elements). More recently, molecular materials have emerged as an intriguing alternative to traditional ferroelectrics and ME materials[10,14–18]. These molecular materials can possess the advantages of both organic and inorganic species within a single phase[19,20]. Moreover, they can be readily modified and optimized by manipulating their specific functional groups[14,21,22]. However, it is important to acknowledge the limitations of molecular ferroelectric and ferromagnetic materials. These drawbacks manifest in the form of

relatively low ordering temperatures for electric or magnetic properties, relatively weak spontaneous polarization, and low melting points, particularly the ME coupling at very low temperatures[23–26] (below the threshold of liquid nitrogen). So far, it remains a great challenge to assemble molecular material with ME coupling at room temperature and low field.

In the ME coupling research, the main attention is focused on those multiferroics with the coexistence of ferroelectric and ferromagnetic orders. However, it is extremely challenging to obtain such coexistence within single-phase molecule-based materials because molecular complexes are difficult to be magnetically ordered. In 2020, Long et al. reported a molecular ME coupling material, [Zn(OAc)(L) Yb(NO₃)₂] (H₂L = 6,6'-((1E,1'E)-((1,2-diphenylethane-1,2-diyl)bis(azaneylylidene))-bis(methaneylylidene))bis(2-methoxyphenol)), which can exhibit switching properties at room temperature and low field[24]. This pioneering study found two important facts. The first is that an effective ME coupling can be obtained even in the paramagnetic phase without long-range magnetic ordering. The second is that magnetostriction can be a good approach to demonstrate the ME coupling in ferroelectrics. In magnetostrictive materials, the applied magnetic field causes the structural deformation of the material by changing the bond lengths and bond angles, which inevitably leads to the change of internal dipoles. Therefore, any molecular ferroelectric with magnetostriction phenomenon is a potential ME coupling material. Thus, it will be a convenient way to search ME coupling materials within those complexes with magnetostriction phenomenon. In fact, the magnetic interactions of transition metal ions with unpaired $d$ electrons are generally sensitive to the deformation of the coordination environment, e.g., bond lengths and angles. In principle, these effects can be more significant in molecular ferroelectrics compared to the inorganic ME coupling materials, partially due to their much softer crystalline frameworks.

Following this scenario, we focus on the $[MX_4]^{n-}$ complexes ($M$ = transition metal; $X$ = halide ion) which are prone to ferroelectric properties and found to show the magnetostrictive effect. Herein, we report a Fe(III)-based molecular complex, (TMCM)[FeCl₄] (TMCM = trimethylchloromethylammonium)[27], which exhibits room-temperature robust ME coupling in its paramagnetic-ferroelectric phase with a tensor magnitude reaching as high as 89 mV Oe⁻¹ cm⁻¹.

## Results

### Characterization of ferroelectricity

The structural analysis of the molecular crystal (TMCM)[FeCl₄][27] reveals that it adopts a monoclinic structure at room temperature, belonging to a polar point group of $m$ and a space group of $C_m$ (see the uploaded CIF data acquired at 296 K). Figure 1a shows a unit cell of (TMCM)[FeCl₄]. The [FeCl₄]⁻ anions occupy the eight vertices and the body center, while the TMCM cations reside in the cavities surrounded by these [FeCl₄]⁻ anions. The C–Cl bonds are longer compared to the C–H bonds (Supplementary Fig. 1), resulting in a noticeable decrease in the sphericity of the halogenated TMCM cations. Consequently, the original centrosymmetric structure of (TMCM)[FeCl₄] undergoes a transition to a polar structure due to the modulation of sphericity in the organic cations. In order to further investigate this structural phase transition, temperature dependence of second harmonic generation (SHG) was conducted. As shown in Fig. 1b, the SHG measurements reveal active behavior with distinct signals below 320 K, confirming a transition from a non-centrosymmetric structure to a centrosymmetric one of (TMCM)[FeCl₄]. The SHG signal changes sharply around 320 K. Such a behavior, with a pair of thermal anomaly peaks (321.5 K) in the DSC curves on cooling and heating (Supplementary Fig. 2), suggests a discontinuous first-order phase transition.

Typical polarization–electric field ($P$–$E$) hysteresis loops show that the saturated polarization reaches up to 0.32 μC cm⁻² along the $a$-axis (Fig. 1c) and 6.1 μC cm⁻² along the $c$-axis (Supplementary Fig. 3), which is the direct evidence of ferroelectricity[28,29]. Further, the local ferroelectric domains within (TMCM)[FeCl₄] were also investigated by piezoelectric force microscopy (PFM). By using PFM (DART-SS-PFM mode, Asylum Research), systematic research of the piezoelectric

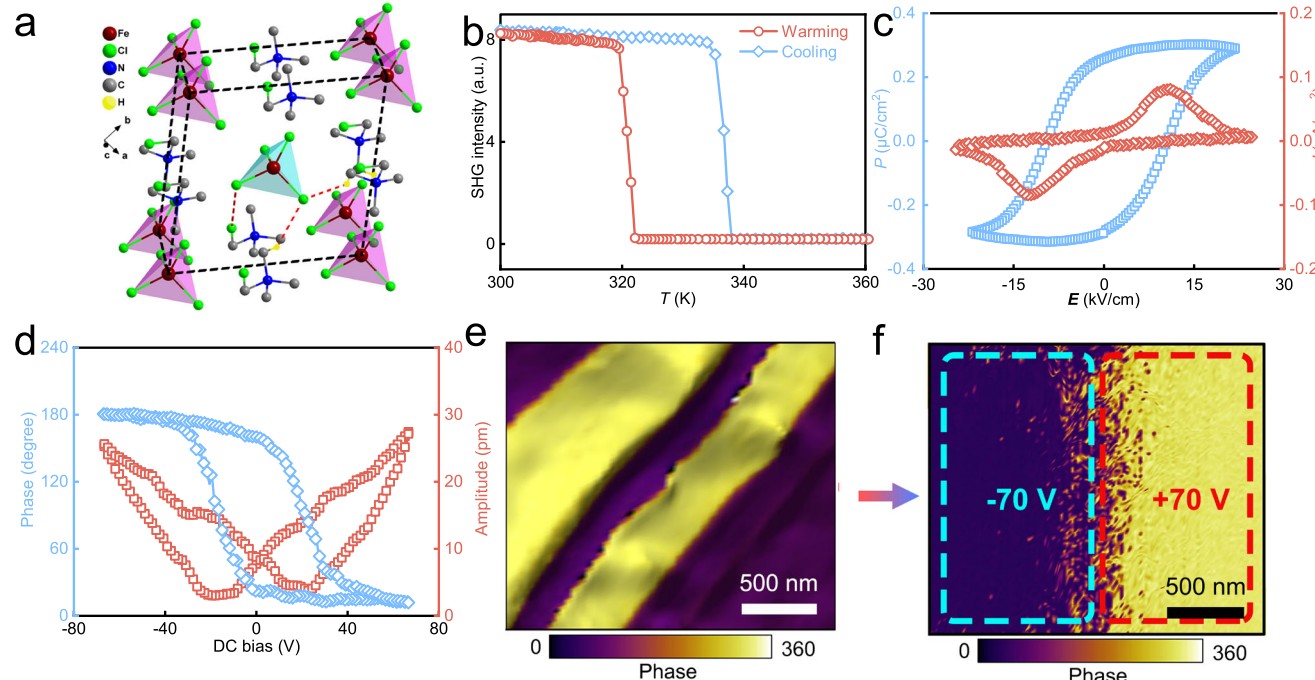

**Fig. 1 | Characterization of ferroelectricity along the $a$-axis. a** Structure of molecular crystal (TMCM)[FeCl₄] at 296 K. **b** Temperature dependence of second harmonic generation (SHG) response. **c** $P$–$E$ (blue) and $J$–$E$ (red) hysteresis loop of (TMCM)[FeCl₄], measured at 310 K. **d** Phase-voltage hysteresis (blue) and amplitude-voltage butterfly loop (red) of the TMCM[FeCl₄] acquired by PFM. **e** Vertical PFM phase image overall morphology showing spike-like ferroelectric domain. **f** Domain reversal of TMCM[FeCl₄] after electrical poling with the DC bias applied (±70 V).

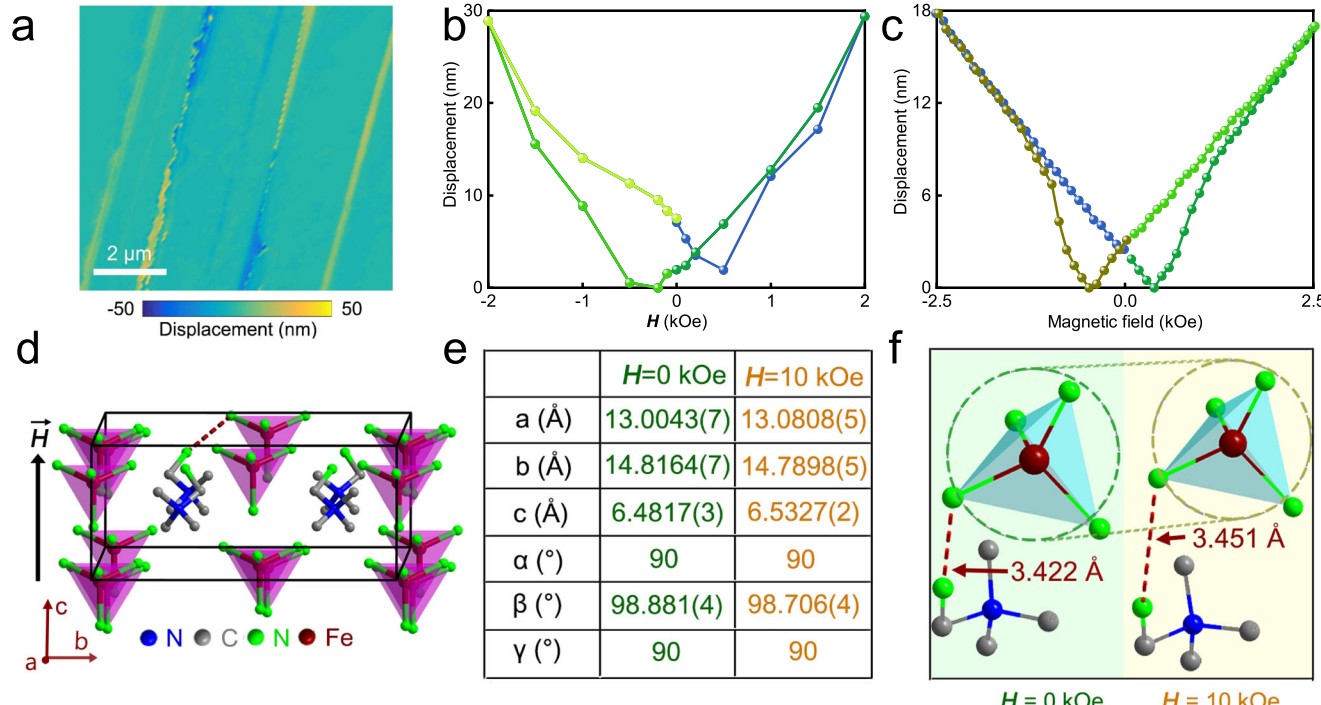

**Fig. 2 | Characterization of magnetostriction. a** In situ displacement of AFM morphology in (TMCM)[FeCl₄] under parallel magnetic field of 2 kOe. **b** Magnetostriction loop measured on a single crystal from −2 to +2 kOe acquired by AFM. The lines are guides for the eye. **c** Magnetostriction as a function of the magnetic field at room temperature measured by PPMS. **d** Illustration of the applied magnetics along the *c*-axis of (TMCM)[FeCl₄]. **e** Structural data for (TMCM)[FeCl₄] obtained at zero and under applied magnetic field. (TMCM)[FeCl₄] becomes lengthening in in *ac* plane while it becomes shorter *b*-axis. **f** The detailed structural deformation in (TMCM)[FeCl₄].

measurements was performed, as depicted in Supplementary Fig. 4. The linearity serves as the indication of the intrinsic piezoelectricity in the two directions (both *a*- and *c*-axis). Subsequently, the switching spectroscopy piezoresponse force microscopy (SS-PFM) measurements in OFF-field mode were carried out. As shown in Fig. 1d, once the applied electric field exceeds the coercive field, the phase presents 180° switching, and the amplitude hysteresis loop shows a butterfly-shaped curve, demonstrating switchable spontaneous polarization along the *a*-axis. Furthermore, Fig. 1e provides a vertical PFM phase image overall the morphology of the (100) plane. The ferroelectric domains manifest as spike-like patterns that reflect distinct polarization orientations[30]. The PFM amplitude images acquired simultaneously are shown in Supplementary Fig. 5 implying the relative magnitudes of ferroelectric domains. The ferroelectric properties for the *c*-axis were almost similar to those for the *a*-axis, as shown in Supplementary Fig. 6. The PFM characteristics and local domains further confirm the ferroelectricity along the *c*-axis. Finally, we evaluated the reversal of ferroelectric domains after electrical poling. The 'up' and 'down' oriented domains are visualized in purple and orange, respectively, along the *a*-axis (Fig. 1f), thereby confirming the intrinsic ferroelectric nature in (TMCM)[FeCl₄].

### Characterization of magnetostriction

Polycrystalline samples of (TMCM)[FeCl₄] were subjected to static direct-current susceptibilities ($\chi_M T$) measurements in an applied field of 1 kOe across a temperature range of 2.0–370 K (Supplementary Fig. 7). At room temperature, the $\chi_M T$ value is 4.401 cm³ mol⁻¹ K, very close to (only slightly higher than) the expected value of 4.375 cm³ mol⁻¹ K for single Fe(III) ions (spin $S = 5/2$ and Lande factor $g = 2$). Interestingly, the analysis of the plot of $\chi_M T$ versus temperature in the range of 300–370 K reveals a transition occurring at approximately 323 K in (TMCM)[FeCl₄] (inside of Supplementary Fig. 7). Since

it occurs in the paramagnetic region, it can not be attributed to the magnetic phase transition. Instead, it should be associated with the ferroelectric structural phase transition, which can also affect the magnetic susceptibilities subtly[14,27].

The magnetostriction of (TMCM)[FeCl₄] crystals was further investigated using atomic force microscopy (AFM) and a physical property measurement system (PPMS). Figure 2a displays the vertical displacement along the *a*-axis observed by AFM under an in-plane magnetic field of 2 kOe. It is observed that most regions of (TMCM)[FeCl₄] exhibit an increase in thickness of approximately 10 nm along the *a*-axis. However, different regions display nonuniform mechanical deformations in response to the external magnetic field. Some regions exhibit positive magnetostriction, while others show negative magnetostriction, which implies the existence of multiple equal states of magnetostriction under the same magnetic field. Figure 2b illustrates the magnetostriction loop at a specific position, with displacements increasing by approximately $2 \times 10^{-4}$ when subjected to the magnetic field of 2 kOe. Further, these results can be confirmed by PPMS measurements as shown in Fig. 2c. Magnetostriction loops are observed in the direction of the *a*-axis by reversing the magnetic field in *bc* plane. The magnetostriction reaches a magnitude of $2 \times 10^{-4}$ at 2.5 kOe at room temperature (Supplementary Fig. 8), which is comparable to the results of AFM measurements. The magnitude of the ME tensor component $\alpha_{31} = \frac{\Delta u}{d_{oop}\Delta HD}$, where $D$ is the thickness of the (100)-oriented crystal, $d_{oop}$ is estimating the relative magnitude of the piezoelectric coefficient from the slope of the curves and $\Delta u$ is the change in vertical surface displacement induced by the magnetic field $\Delta H$, reaches approximately 89 mV Oe⁻¹ cm⁻¹, which is much higher than those observed in multiferroic BiFeO₃ bulk (typically ranging from 0.6 to 7 mV Oe⁻¹ cm⁻¹)[12]. The two equal states of magnetostriction under the same magnetic field are also observed. This phenomenon predominantly exhibits a single-ion character, where the associated

deformation can be equivalent in magnitude to that observed in magnetically long-range ordered systems.

In-situ single-crystal X-ray diffraction was employed to investigate the origin of the magnetostriction. The magnetostriction along the $a$-axis reaches approximately $6 \times 10^{-3}$ when a magnetic field of approximately 10 kOe was applied along the $c$-axis (Fig. 2d and Supplementary Table 1). Figure 2e shows the comparison of structural data before and under an external magnetic field, which supports the results of magnetostriction obtained from AFM and PPMS measurements. Moreover, the length of intermolecular Cl1···Cl7 increases from 3.422 Å to 3.451 Å (Fig. 2f), showing the magnetostriction in the atomic level. Meanwhile, the volume of the $[FeCl_4]^-$ anion is found to become slightly smaller under an external magnetic field (Supplementary Tables 2 and 3).

To further understand the aforementioned experimental observations, a first-principles density functional theory (DFT) calculation was performed. To determine the magnetic ground state of (TMCM) $[FeCl_4]$, the four most possible magnetic orders, including the ferromagnetic one and three antiferromagnetic (AFM) ones, are considered (Supplementary Fig. 9). According to our calculation, the energy of G-AFM is slightly lower than those of other three, as summarized in Supplementary Table 4. The average nearest-neighbor magnetic exchange (denoted as $J$) is estimated to be 0.3 meV (for normalized spin $|S| = 1$). Considering the nearly isolated $[FeCl_4]^-$ groups, such a weak $J$ is reasonable, which can establish the G-AFM at a very low temperature. Thus, the experimental paramagnetic state is consistent with our DFT expectation. Since the room-temperature paramagnetic state can not be directly simulated in the DFT calculation, the theoretical value of polarization is estimated based on the structure of the ground state (G-AFM), which leads to 0.31 μC cm⁻² along the $a$-axis and 8.23 μC cm⁻² along the $c$-axis, very close to (and slightly higher than) the experimental values at room temperature (0.32 μC cm⁻² along the $a$-axis (Fig. 1c) and 6.1 μC cm⁻² along the $c$-axis (Supplementary Fig. 3).

The optimized lattice constants are also consistent with the experimental ones, as compared in Supplementary Table 4. The overall slight constriction of DFT values is reasonable considering the thermal expansion effect since the DFT ones are for zero temperature while the experimental ones are measured at room temperature. Although the paramagnetic state can not be directly mimicked in our DFT calculation, the magnetostrictive effect can be qualitatively simulated. For reference, the variation of ferromagnetic lattice constants ($a$, $b$, $c$) are different from those of G-AFM by 0.15%, −0.036%, and −0.083%, respectively, suggesting a moderate magnetostrictive effect. In fact, the weak magnetic coupling $J$ allows the sensitive paramagnetic response to the external magnetic field. Consequently, the lattice constants are expected to change upon this magnetization process, as observed in experiments. Supplementary Fig. 10 shows its band structure and atomic-projected density of states (DOS), which suggests a moderate band gap (2.2 eV).

## ME coupling behavior

To investigate the ME coupling effect in (TMCM)$[FeCl_4]$, we conducted in situ PFM measurements along the $a$-axis under an in-plane magnetic field of ±2 kOe. The results reveal a redistribution of the ferroelectric domain in Fig. 3a−c, similar to that poled by an electrical bias voltage. The relative magnitudes of ferroelectric domains are captured in the PFM amplitude images shown in Supplementary Fig. 11. This indicates that the change in ferroelectric domain facilitates the observation of the ME coupling effect along the $a$-axis in (TMCM)$[FeCl_4]$. In addition, we observed the resonance and amplitude-drive voltage curves under different magnetic fields. Both the peak and $d_{oop}$ change under the magnetic field (Fig. 3d), and these curves all show good linearity, giving evidence of intrinsic piezoelectric. The slope at ±2 kOe is 50%, smaller than that of (TMCM)$[FeCl_4]$ at 0 Oe, confirming the ME coupling. These remarkable characteristics of (TMCM)$[FeCl_4]$ make it a promising candidate in comparison to traditional metal oxides for various applications in the field of magnetoelectronics. Furthermore, the external magnetic field strongly affects the ferroelectric loops, leading to changes in their symmetry, amplitude, and coercive fields, as depicted in Fig. 3e, and Supplementary Figs. 12 and 13. Ferroelectric

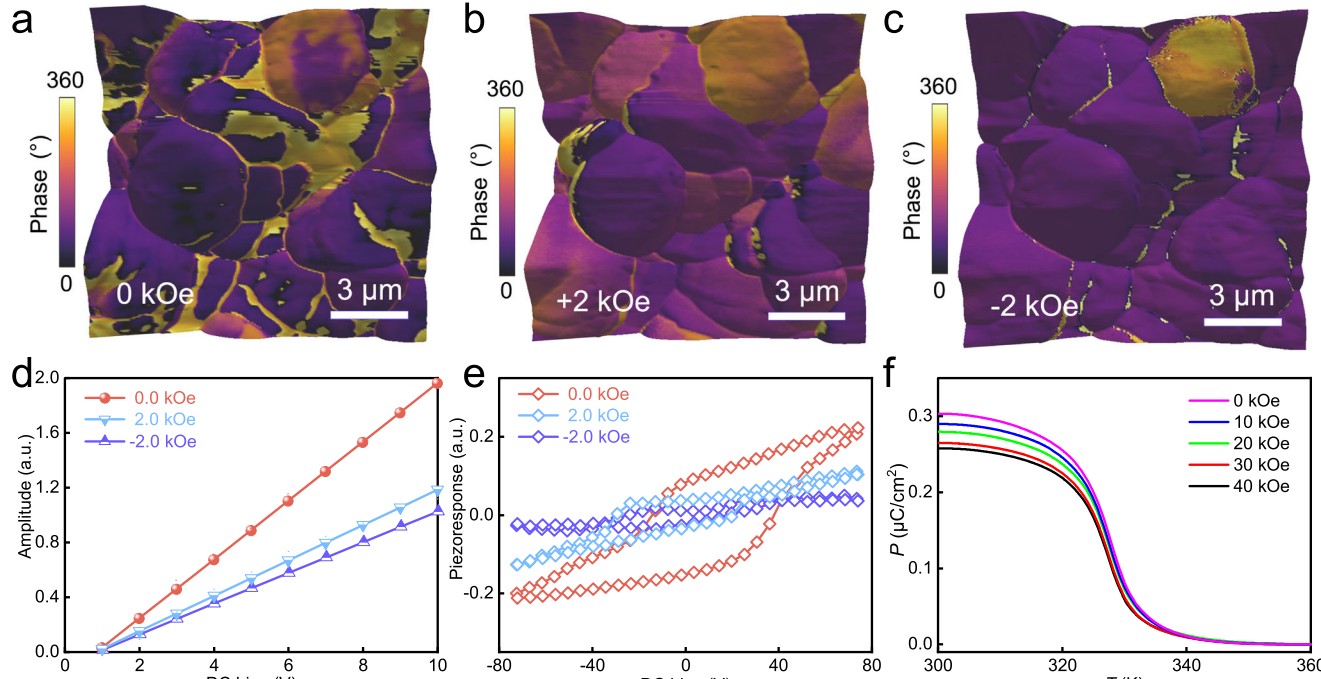

**Fig. 3 | ME coupling behavior. a−c** Vertical PFM ferroelectric domains under in-plane $H = 0$ Oe (**a**), 2 kOe (**b**), and −2 kOe (**c**) underlying the ME coupling effect. **d** The effective piezoelectric coefficient under magnetic fields of 0 Oe and ±2 kOe. **e** (TMCM)$[FeCl_4]$-SS-PFM-OFF-field hysteresis loops obtained at zero and under an applied magnetic field of ±2 kOe. **f** Temperature dependence of $P$ along the $a$-axis under external magnetic fields from 0 to 40 kOe.

polarization along the *a*-axis is suppressed by the magnetic field, which corresponds to the stretching of the halogen Cl···Cl bonds and the shrinking volume occupancy of [FeCl$_4$]$^-$ anions[21].

Furthermore, the temperature-dependent ferroelectric polarization along the *a*-axis was measured under different magnetic fields. As shown in Fig. 3f, the saturated polarization (*P*) along the *a*-axis decreases with increasing magnetic field. To quantify the magnetoelectric coupling, the relative change of polarization is calculated as $\Delta P/P_0 = (P_0 - P_H)/P_0$, where $P_0$ and $P_H$ are the ferroelectric polarizations along the *a*-axis without and with an external magnetic field, respectively. $\Delta P/P_0$ is considerable: ~7% when the magnetic field is 10 kOe and ~12% under the 40 kOe field. These findings provide solid evidence of a pronounced ME coupling effect in TMCM[FeCl$_4$] at room temperature. In addition, the first derivative of polarization of (TMCM)[FeCl$_4$] exhibits a dominant negative dip (Supplementary Fig. 14). The position of the local dip shifts to lower temperature with increasing magnetic fields, which suggests that (TMCM)[FeCl$_4$] undergoes a first-order phase transition. This finding is in good agreement with the DSC analysis.

## Outlook

We have demonstrated the room-temperature ME coupling under low magnetic fields in a molecular ferroelectric material TMCM[FeCl$_4$]. The combination of ferroelectricity and magnetostriction effect leads to the robust ME coupling effect. The effect is induced by the stretching of halogen Cl···Cl contacting and the change in volume occupancy of [FeCl$_4$]$^-$ anions, which can be finely tuned under low magnetic fields. These interactions result in the interplay between the magnetic and electric properties, leading to the ME coupling effect. This characteristic is significant for practical application as it allows for the manipulation in low fields at room temperature. This ME effect induced by magnetostriction opens up a possibility for exploring new materials and further expanding the variety of systems that can harness the benefits of ME coupling.

## Methods
### DSC, SHG, PXRD, and single-crystal measurements

Differential scanning calorimetry (DSC) was measured using NETZSCH DSC (214 Polyma), with nitrogen atmosphere protection in aluminum crucibles. The heating/cooling rate during the DSC measurements is 10 K/min. A pump Nd:YAG laser (1064 nm, 1 Hz repetition rate) was used to measure the SHG intensity on powder samples (75–150 μm) in the temperature range from 300 to 338 K. Power X-ray diffraction (PXRD) was measured using a Bruker D8 Advance diffractometer (Cu Kα radiation $\lambda = 1.54184$ Å), from 298 K to 333 K. The PXRD patterns were scanned in the 5° <2θ < 40° range with a step size of 0.019° within a period of 1.5 h. Single crystals of (TMCM)[FeCl$_4$] were observed directly with the solution evaporates. The variable temperature single crystal data of compound (TMCM)[FeCl4] were collected under a liquid nitrogen atmosphere on a Bruker D8 Venture. The single crystal structure under zero magnetic field and the single crystal structure after stimulated by 10 kOe magnetic field were measured at room temperature.

### Scanning electron microscopy (SEM), transmission electron microscopy (TEM), atomic force microscopy (AFM), piezoelectric force microscopy (PFM), and ferroelectric measurements

The samples were observed on a 5350 NE Dawson Creek Derive SEM. The samples of TEM were measurement on JEM−200CX. PFM and AFM were conducted at the same time and located on a NanoScope IV NS4-1 instrument. During the AFM, PFM measurements the height sensor below 1 μm. *J*–*V* curves were measured using the double-wave method, which can remove non-hysteresis components in *P*–*E* loops[31].

### Magnetic characteristics

The direct current (dc) magnetic data of (TMCM)[FeCl4] were collected under 1.0 kOe with temperature range of 2–370 K.

### Magnetoelectric characteristics

The vertical displacement versus parallel magnetic field relationship of the molecular materials TMCM(FeCl$_4$) was measured at different temperatures on a PPMS-9 of Quantum Design. AFM was conducted at the same time under different parallel magnetic fields in a NanoScope IV NS4-1 instrument. During the AFM measurements, the height sensor was below 1 μm.

### DFT calculation

The density functional theory (DFT) calculations were performed with projector-augmented wave pseudopotentials as implemented in the Vienna ab initio Simulation Package (VASP)[32]. For the exchange-correlation functional, the Perdew–Burke–Ernzerhof parameterization of the generalized gradient approximation (GGA) was adopted[33], and the Hubbard *U* was applied using the Dudarev parameterization[34]. $U_{eff} = 4$ eV for Fe's 3*d* orbitals were employed[35]. The energy cutoff was fixed to 500 eV, and the *k*-point grids of $2 \times 2 \times 4$ were adopted for both optimization and static calculation. The convergent criterion for the energy was set to $10^{-6}$ eV, and the criterion of the Hellman–Feynman force during the structural relaxation was 0.01 eV/Å for all atoms. In addition, the vdW correction of the DFT-D3 method was adopted[36]. The polarization was calculated using the standard Berry phase method by Eq. 1[37,38]:

$$\mathbf{P} = \mathbf{P_e} + \mathbf{P_{ion}} = \frac{e}{(2\pi)^3} \operatorname{Im} \sum_n \int d\mathbf{k} <u_{nk}|\nabla_k|u_{nk}> + \frac{e}{\Omega} \sum_s Z_s^{ion} \mathbf{r}_s \quad (1)$$

where the first part $\mathbf{P_e}$ denotes the valence electronic contribution calculated by the Berry phase method. $u_{nk}$ is the wave function, the integrated interval is done over the Brillouin zone, and the sum *n* runs over all occupied bands. The second term, $\mathbf{P_{ion}}$, denotes the ionic contribution. *e*: elementary charge; Ω: volume of unit cell; $Z^{ion}$: charge of ion, and **r**: position.

### Reporting summary

Further information on research design is available in the Nature Portfolio Reporting Summary linked to this article.

## Data availability

All data generated and analyzed in this study are included in the article and Supplementary Information and are also available at the corresponding authors' request. The crystal structures generated in this study have been deposited in the Cambridge Crystallographic Data Center under accession code CCDC: 2310264−2310266 and can be obtained free of charge from the CCCDC via www.ccdc.cam.ac.uk/data_request/cif.

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

## Acknowledgements

This work is supported by the Joint Fund for Regional Innovation and Development (U20A2073), the National Natural Science Foundation of China (22375182, 92056112, 22373048, 62004136, 22105089, and 12325401), Natural Science Foundation of Jiangxi Province (20224BAB214005), the Natural Science Foundation of Zhejiang Province (LZ24B010001), the Interdisciplinary Program of Wuhan National High Magnetic Field Center (WHMFC202133) and the Big Data Center of Southeast University for providing the facility support on the numerical calculations.

## Author contributions

Z.B.H., Y.Z., and Y.S. conceived the project. S.D. proposed the theoretical scenario. L.A.G and Y.F.Z. designed the experiments. X.D.L. and Z.H.Z. measured and analyzed the magnetic properties of (TMCM) [FeCl4]. Y.J. measured the P–E curves. Z.B.H. prepared the samples and performed the DSC, SHG, PFM, magnetoelectricity, and ME coupling measurements. X.Y. performed the DFT calculations guided by S.D. J.L.Z. contributed to the analysis of PFM. S.D., Y.Z., and Y.S. wrote the paper, with inputs from all other authors. Z.B.H. X.Y., and J.L.Z. contributed equally.

## Competing interests

The authors declare no competing interests.

## Additional information

[1]State Key Laboratory of Coordination Chemistry, School of Chemistry and Chemical Engineering, Nanjing University, Nanjing 210023, China. [2]Chaotic Matter Science Research Center, Department of Materials, Metallurgy and Chemistry & Jiangxi Provincial Key Laboratory of Functional Molecular Materials Chemistry, Jiangxi University of Science and Technology, Ganzhou 341000, China. [3]Key Laboratory of Quantum Materials and Devices of Ministry of Education, School of Physics, Southeast University, Nanjing 211189, China. [4]Advanced Technology Research Institute of Taihu Photon Center, Suzhou University of Science and Technology, Suzhou 215009, China. [5]Institute for Science and Applications of Molecular Ferroelectrics, Key Laboratory of the Ministry of Education for Advanced Catalysis Materials, Zhejiang Normal University, Jinhua 321004, China.
✉e-mail: zhangjinlei@usts.edu.cn; yizhang1980@seu.edu.cn; sdong@seu.edu.cn; yousong@nju.edu.cn

