## [Peer Review File · Nature Communications]

Molecular ferroelectric with low-magnetic-field magnetoelectricity at room temperatureEditorial Note: Parts of this Peer Review File have been redacted as indicated to remove third-party material where no permission to publish could be obtained.

REVIEWER COMMENTS

Reviewer #1 (Remarks to the Author):

The authors constructed a single phase ferroelectric molecule of [TMCM][FeCl₄], which was another new molecular ferroelectric different from the molecular ferroelectric ytterbium(III) complex [Long. J. et.al., Room temperature magnetoelectric coupling in a molecular ferroelectric ytterbium(III) complex, *Science*, 367, 671-676 (2020)]. Its room-temperature magnetoelectric (ME) coupling was demonstrated at low magnetic fields. They revealed that the magnetostriction effect induced by the in-plane magnetic field was responsible for the robust ME coupling. This magnetostriction effect manifested the structural deformation with the stretching of halogen Cl...Cl contactings and the change in volume occupancy of [FeCl₄]⁻ anions at low magnetic fields. Furthermore, the magnetostriction effect is qualitatively simulated by density functional theory (DFT) calculations. The methodology seems to be sound. The findings presented in the manuscript are overall scientifically sound and supported by the data and detailed analysis, I recommend publication of the work in *Nature communications* after the below concerns are addressed.

1. From Fig. S2, the authors claimed that the phase transition with a pair of thermal anomaly peaks was observed at temperatures of 321.5/338.1 K on cooling and heating, suggesting a discontinuous first-order phase transition. Nevertheless, from Fig. 1(b) and Fig. 3(f), one could find that the SHG intensity and ferroelectric polarization dropped to zero continuously, respectively, indicating the second-order phase transition. Could the authors provide clarification on this?

2. Along c-axis, the polarization-electric field (P-E) hysteresis loop with saturated polarization (PS) reaching up to 6 $\mu\text{C cm}^{-2}$, which was mentioned in the text, was not found in Fig. 1(c).

3. In Fig. S5, it was shown that the χ_{MT} value was 4.79 $\text{cm}^3 \text{mol}^{-1} \text{K}$, slightly higher than the expected value of 4.375 $\text{cm}^3 \text{mol}^{-1} \text{K}$ for single Fe(III) ions ($S = 5/2$ and $g = 2.0$). The authors claimed that this difference in susceptibility was primarily attributed to orbital contributions resulting from spin-orbit coupling. If the dissociative [FeCl₄]⁻ anion as magnetic impurity existed in single crystal, it may also result in slightly higher paramagnetic value of susceptibility. Thus, the purity of single crystal and effective moment should be further tested to identify the orbital contributions resulting from spin-orbit coupling.

4. In Fig. S5, the curve of χ_{MT} versus T increases with temperature ascending, and then keeps a constant, implying the low-temperature antiferromagnetism crossover into high-temperature paramagnetism. The antiferromagnetism has been further identified by the DFT calculation. Around 323K, the authors claimed the occurrence of paramagnetic phase transition. It may be a ferroelectric transition accompanied by structural transition rather than a magnetic phase transition. May the authors please clarify? Furthermore, at low temperatures, the antiferromagnetic order and ferroelectric order coexist, i.e., multiferroics. Is there any intrinsic ME coupling at low temperatures, compared to the room temperature paramagnetic ME coupling induced by low magnetic fields?

5. Fig. 1(d) was still labelled as Fig. 1(c). It should be revised.

Reviewer #2 (Remarks to the Author):

MANUSCRIPT

Molecular ferroelectric with low-magnetic-field magnetoelectricity at room temperature

The authors investigated the paramagnetic ferroelectric molecule i.e. [TMCM][FeCl₄] shows the coupled magnetic and electric polarizabilities, providing multiple high quality experimental evidence to support the claim along with theoretical results obtained from DFT calculations. Highlighting the stretching of the Cl-Cl bonds and change sphericity of the molecules leads to the robust ME coupling effect. It is commendable to see these exotic properties at the room temperature coupled with a low-magnetic-field in this type of molecule different from the traditional inorganic ferroelectric materials.

However, at few points adding more DFT or theoretical evidence will highly elevate the impact of the current work, this manuscript could be considered for publication once the following issues has been addressed.

[1] Author had evidence from the DFT calculations that support the experimental results such as lattice parameters and magnetic moment. However, it would be highly recommended to calculate the ferroelectric polarization value for this molecule using DFT. It won't be as trivial in comparison to the paramagnetic states.

[2] Fig. 1 labels were not in order, also the information such as on line 107 "6 $\mu\text{C cm}^{-2}$ along c-axis" not shown anywhere in the plots.

[3] Author should consider some good color scheme to plot the molecule, difficult to distinguish H atoms.

[4] "The ferroelectric domains manifest as spike-like patterns that reflect distinct polarization orientations", author should acknowledge the prior theoretical work citing the Carbon Trends 11 (2023): 100264, showing the significance of the domain wall roughness associated with the FE-domains in the traditional ABO₃ material.

[5] On line 133, author mentioned "This transition can be associated with changes in orbital contributions related to the structural phase transition experienced by the material", please provide the appropriate evidence or prior work to support this claim.

[6] Similarly, on line 164, please provide the supporting evidence (theoretical or experimental) for the claim "could also lead to a decrease in the energy barrier of cationic motion, which may decrease the ferroelectric polarization."

[7] Author should consider rewriting this sentence on line 211 "volume occupancy of [FeCl₄]⁻ anions", what happens to volume occupancy?

[8] Author should specify the meaning or reason showing different colors in the Supplementary Figure 6?

[9] Author should rewrite the explanation of figure 1d on line 115-116

Reviewer #3 (Remarks to the Author):

In this work, the authors claim proposing a methodology for constructing single phase room-temperature ME ferroelectrics by harnessing magnetostriction via spin-orbit coupling. Although the subject of magnetoelectricity is intriguing, most of the conclusions of the different techniques remain vague, the outcomes of the manuscript are in my opinion seems contradictory

I can not recommend the acceptance due to following reasons.

-- The overall presentation is poor, falls short of the high standards expected by Nature Communications. The text is marred by numerous errors, such as an incorrect icon in fig. 1 and a misalignment between the content depicted in fig. 1c and its corresponding text description in the main text. Several images are typically not standardized.

-- The authors claim proposing a methodology for constructing single phase room-temperature ME ferroelectrics by leveraging magnetostriction via spin-orbit coupling. While magnetostrictive mechanisms in magnetoelectric materials have been documented, it is important to note that such mechanisms likely exist in various ME materials with differing contributions. Consequently, the claims regarding the methodology have been significantly overstated.

-- The compound under investigation does not seem to be a new material. Without diving too deep into the literature, also found that significant overlap with characterization data found in reference 27 provided by the authors.

-- The paper contains numerous instances of contradictory data, such as:

- In fig. 1c, a substantial polarization value is detected at 323 K, while DSC data contradicts this observation by indicating the compound is in the paraelectric phase at this temperature. the authors demonstrate a significant polarization value detected at 323 K. Even more confusing, the data in fig. 3f indicates that the polarization value at 323 K is inexplicably 0.
- The magnetostriction results obtained by In-situ single-crystal X-ray diffraction also appear inconsistent with theoretical calculation results.

-- Many data provided by the author appear Inconsistencies. For instance, in fig. 1f, the display indicates 70 V, yet the accompanying icons suggest 20 V. Supplementary Table 1 present the values along axis 'a' with clear elongation, in contrast to the author's values, which appear shortened. In summary, the dataset appears messy and inconsistent.

Considering these issues, this manuscript fails to meet the standards required for publication in a prestigious journal such as Nature Communications.

Response to Reviewer #1

General Comment: The authors constructed a single phase ferroelectric molecule of [TMCM][FeCl₄], which was another new molecular ferroelectric different from the molecular ferroelectric ytterbium(III) complex [Long. J. et.al., Room temperature magnetoelectric coupling in a molecular ferroelectric ytterbium(III) complex, *Science*, 367, 671-676 (2020)]. Its room-temperature magnetoelectric (ME) coupling was demonstrated at low magnetic fields. They revealed that the magnetostriction effect induced by the in-plane magnetic field was responsible for the robust ME coupling. This magnetostriction effect manifested the structural deformation with the stretching of halogen Cl...Cl contactings and the change in volume occupancy of [FeCl₄]- anions at low magnetic fields. Furthermore, the magnetostriction effect is qualitatively simulated by density functional theory (DFT) calculations. The methodology seems to be sound. The findings presented in the manuscript are overall scientifically sound and supported by the data and detailed analysis, I recommend publication of the work in *Nature communications* after the below concerns are addressed.

Response: We thank the reviewer for his/her careful reading, precise summary, and recommendation.

Comment 1: From Fig. S2, the authors claimed that the phase transition with a pair of thermal anomaly peaks was observed at temperatures of 321.5/338.1 K on cooling and heating, suggesting a discontinuous first-order phase transition. Nevertheless, from Fig. 1(b) and Fig. 3(f), one could find that the SHG intensity and ferroelectric polarization dropped to zero continuously, respectively, indicating the second-order phase transition. Could the authors provide clarification on this?

Response: Many thanks for this suggestion. In general, the type of phase transition (first-order or second-order) for ferroelectric materials can be determined by DSC, SHG, and pyroelectric curves. There are obvious differences in DSC curves between the first-order and second-order transitions. The first-order phase transition has larger thermal entropy change and thermal hysteresis (e.g. Fig. R1a) compared with those of second-order phase transition (e.g. Fig. R1c). The DSC curve of (TMCM)[FeCl₄] is close to the first-order phase transition (see Fig. R2a). Moreover, the temperature dependent SHG strength are also significantly different. For the first-order phase transition, the SHG strength changes less with temperature below T_C and presents a discontinuous sharp decrease to 0 at (near) T_C , as shown in Fig. R1b. For the second-order phase transition, the SHG intensity presents a continuous decrease with temperature, as shown in Fig. R1d.

[Redacted]

Fig. R1. Characteristics of ferroelectric phase transitions. a-b. First-order transition. Reproduced from *Science* 361, 151 (2018). c-d. Second-order transition. Reproduced from *J. Am. Chem. Soc.* 137, 4928 (2015).

Fig. R2. Characteristics of our ferroelectric transition, which is obviously the first-order one.

The temperature dependence of SHG intensity of our (TMCM)[FeCl₄] sample tends to be a first-order phase transition (Fig. R2b). Furthermore, the temperature-dependent pyroelectric curves for the first-order and second-order transitions are also significantly different. The polarization (P) varies less with temperature for the first-order phase transition before a discontinuous sharp decrease of polarization to 0

near T_C , as shown in Fig. R1b. For the second-order phase transition, the P - T curve presents relatively uniform and continuous decrease over a large temperature range (e.g. ~ 50 K) till T_C , as shown in Fig. R1e. The polarization of (TMCM)[FeCl₄] decreases sharply from a large value to 0 in a very narrow T -range (~ 5 K) under different magnetic fields (Fig. 3(f)). Therefore, all these results indicate that the phase transition of (TMCM)[FeCl₄] is likely to be a first-order one.

Corresponding revision: Lines 92-95 are revised as “The SHG signal changes sharply around 320 K and becomes less-changed below 315 K. Such a behavior, with a pair of thermal anomaly peaks (321.5 K and 338.1 K) in the DSC curves on cooling and heating (Supplementary Fig. 2), suggesting a discontinuous first-order phase transition.”

Comment 2: Along c -axis, the polarization-electric field (P - E) hysteresis loop with saturated polarization (PS) reaching up to $6 \mu\text{C cm}^{-2}$, which was mentioned in the text, was not found in Fig. 1(c).

Response: Thank you very much for the reviewer's reminder. We have corrected it and added the P - E curve in the revised Supporting Information (Fig. S3).

The corresponding revision: “Typical polarization-electric field (P - E) hysteresis loops show that the saturated polarization (PS) reaches up to $0.3 \mu\text{C cm}^{-2}$ along the a -axis (Fig. 1c) and $6 \mu\text{C cm}^{-2}$ along the c -axis (Fig. S3).”

Fig. S3.

Comment 3: In Fig. S5, it was shown that the $\chi_M T$ value was $4.79 \text{ cm}^3 \text{ mol}^{-1} \text{ K}$, slightly higher than the expected value of $4.375 \text{ cm}^3 \text{ mol}^{-1} \text{ K}$ for single Fe(III) ions ($S = 5/2$ and $g = 2.0$). The authors claimed that this difference in susceptibility was primarily attributed to orbital contributions resulting from spin-orbit coupling. If the dissociative $[\text{FeCl}_4]^-$ anion as magnetic impurity existed in single crystal, it maybe also result in slightly higher paramagnetic value of susceptibility. Thus, the purity of single crystal and effective moment should be further tested to identify the orbital contributions resulting from spin-orbit coupling.

Response: It is a good question. We have again collected direct-current magnetic data of complex 1.

To ensure the purity of the sample, we measured the magnetic properties of powder sample from a big crystal. This process can exclude possible magnetic impurity attached at the surfaces of multiple crystals. The result has been added in the revised Supporting Information (Fig. S6). Now the deviation between the experimental value and expected value has been reduced from 9.5% to 0.6%, implying a very nice agreement.

The corresponding revision: “At room temperature, the $\chi_M T$ value is $4.401 \text{ cm}^3 \text{ mol}^{-1} \text{ K}$, very close to (slightly higher than) the expected value of $4.375 \text{ cm}^3 \text{ mol}^{-1} \text{ K}$ for single Fe(III) ions ($S = 5/2$ and $g = 2.0$).”

Comment 4: In Fig. S5, the curve of $\chi_M T$ versus T increases with temperature ascending, and then keeps a constant, implying the low-temperature antiferromagnetism crossover into high-temperature paramagnetism. The antiferromagnetism has been further identified by the DFT calculation. Around 323K, the authors claimed the occurrence of paramagnetic phase transition. It may be a ferroelectric transition accompanied by structural transition rather than a magnetic phase transition. May the authors please clarify? Furthermore, at low temperatures, the antiferromagnetic order and ferroelectric order coexist, i.e., multiferroics. Is there any intrinsic ME coupling at low temperatures, compared to the room temperature paramagnetic ME coupling induced by low magnetic fields?

Response: The reviewer’s opinion is correct. It is paramagnetic at high temperature. The anomaly of $\chi_M T$ at 323 K was owing to the structural transition. Indeed, there are intrinsic ME couplings at low temperatures. Unluckily, in our laboratory, we do not have instruments to characterize the ME coupling at low temperatures, therefore we did not discuss it in this work. If suitable characterization methods are available in the future, we will try to study multiferroics behavior at low temperatures. Even though, it does not affect our conclusion on room-temperature ME behavior, which is even more interesting for potential applications than the low-temperature one.

Comment 5: Fig. 1(d) was still labelled as Fig. 1(c). It should be revised.

Response: Thanks for pointing out this typo. We have corrected it.

Response to Reviewer #2

General Comment: The authors investigated the paramagnetic ferroelectric molecule i.e. [TMCM][FeCl₄] shows the coupled magnetic and electric polarizabilities, providing multiple high quality experimental evidence to support the claim along with theoretical results obtained from DFT calculations. Highlighting the stretching of the Cl-Cl bonds and change sphericity of the molecules leads to the robust ME coupling effect. It is commendable to see these exotic properties at the room temperature coupled with a low-magnetic-field in this type of molecule different from the traditional inorganic ferroelectric materials. However, at few points adding more DFT or theoretical evidence will highly elevate the impact of the current work, this manuscript could be considered for publication once the following issues has been addressed.

Response: We thank the reviewer for his/her careful reading, precise summary, and recommendation.

Comment 1: Author had evidence from the DFT calculations that support the experimental results such as lattice parameters and magnetic moment. However, it would be highly recommended to calculate the ferroelectric polarization value for this molecule using DFT. It won't be as trivial in comparison to the paramagnetic states.

Response: Thanks for this suggestion. In the revised manuscript, we have calculated the ferroelectric polarization for this molecule based on the structure of ground state (G-AFM) for reference, since the room-temperature paramagnetic state can not be directly simulated in the DFT calculation. According to our calculation, the polarization is 0.31 $\mu\text{C cm}^{-2}$ along the *a*-axis and 8.23 $\mu\text{C cm}^{-2}$ along the *c*-axis, very close to (and slightly higher than) the experimental values at room temperature (0.32 $\mu\text{C cm}^{-2}$ along the *a*-axis (Fig. 1c) and 6.1 $\mu\text{C cm}^{-2}$ along the *c*-axis (Fig. S3)).

The corresponding revision: "Since the room-temperature paramagnetic state can not be directly simulated in the DFT calculation, the theoretical value of polarization are estimated based on the structure of ground state (G-AFM), which leads to 0.31 $\mu\text{C cm}^{-2}$ along the *a*-axis and 8.23 $\mu\text{C cm}^{-2}$ along the *c*-axis, very close to (and slightly higher than) the experimental values at room temperature (0.32 $\mu\text{C cm}^{-2}$ along the *a*-axis (Fig. 1c) and 6.1 $\mu\text{C cm}^{-2}$ along the *c*-axis (Fig. S3))."

Comment 2: Fig. 1 labels were not in order, also the information such as on line 107 "6 $\mu\text{C cm}^{-2}$ along *c*-axis" not shown anywhere in the plots.

Response: We have fixed the labels in Fig.1 and added Fig. S3 in the revised Supporting information.

Comment 3: Author should consider some good color scheme to plot the molecule, difficult to distinguish H atoms.

Response: Following this suggestion, we have changed the color scheme, as follows.

Fig. 1a

Comment 4: “The ferroelectric domains manifest as spike-like patterns that reflect distinct polarization orientations”, author should acknowledge the prior theoretical work citing the Carbon Trends 11 (2023): 100264, showing the significance of the domain wall roughness associated with the FE-domains in the traditional ABO_3 material.

Response: Thanks for bringing us attention on this paper. We have cited it as Ref. 30 in the revised manuscript.

Comment 5: On line 133, author mentioned “This transition can be associated with changes in orbital contributions related to the structural phase transition experienced by the material”, please provide the appropriate evidence or prior work to support this claim.

Response: We have added Refs. 14 and 27 in the revised manuscript. And the phenomenon of magnetic change was reported in Ref. 27.

Comment 6: Similarly, on line 164, please provide the supporting evidence (theoretical or experimental) for the claim “could also lead to a decrease in the energy barrier of cationic motion, which may decrease the ferroelectric polarization.”

Response: We have added Ref. 31 in the revised manuscript, which reported the phase transition temperature in complexes $[(CH_3)_3NCH_2X]FeBr_4$ ($X = F, Cl, Br, I$). The strong of $X \cdots H$ bond, the higher the phase transition temperature of compounds. And the ferroelectric polarization is small (see Fig. R3).

[Redacted]

Fig. R3. Comparison of P-E loops of $[(CH_3)_3NCH_2X]FeBr_4$ ($X = F, Cl, Br, I$). Reproduced from Ref. 31

Comment 7: Author should consider rewriting this sentence on line 211 “volume occupancy of $[\text{FeCl}_4]^-$ anions”, what happens to volume occupancy?

Response: Thanks for this suggestion. We have deleted it in the revised manuscript.

Comment 8: Author should specify the meaning or reason showing different colors in the Supplementary Figure 6?

Response: We add the description in the revised Supporting Information as follow: “The olive colours represents the results of the field-up measuring, while orange represents the results of the field-down measuring.”

Comment 9: Author should rewrite the explanation of figure 1d on line 115-116.

Response: Thanks for this suggestion. We have rewritten the explanation of figure 1d in revised manuscript as follow:

“Subsequently, the switching spectroscopy piezoresponse force microscopy (SS-PFM) measurements in OFF-field mode were carried out. As shown in Figs. 1d and S5, when the applied electric field exceeds the coercive field, the phase presents 180° switching, and the amplitude hysteresis loop shows a butterfly-shaped curve, demonstrating switchable spontaneous polarization.”

Response to Reviewer #3

General Comment: In this work, the authors claim proposing a methodology for constructing single phase room-temperature ME ferroelectrics by harnessing magnetostriction via spin-orbit coupling. Although the subject of magnetoelectricity is intriguing, most of the conclusions of the different techniques remain vague, the outcomes of the manuscript are in my opinion seems contradictory. I can not recommend the acceptance due to following reasons.

Response: We thank the reviewer for his/her reading and comments/suggestions. We hope the revised version can change his/her opinion.

Comment 1: The overall presentation is poor, falls short of the high standards expected by Nature Communications. The text is marred by numerous errors, such as an incorrect icon in fig. 1 and a misalignment between the content depicted in fig. 1c and its corresponding text description in the main text. Several images are typically not standardized.

Response: We sincerely apologize for the improper presentation. We have carefully polished full manuscript and corrected all mistakes in revised manuscript.

Comment 2: The authors claim proposing a methodology for constructing single phase room-temperature ME ferroelectrics by leveraging magnetostriction via spin-orbit coupling. While magnetostrictive mechanisms in magnetoelectric materials have been documented, it is important to note that such mechanisms likely exist in various ME materials with differing contributions. Consequently, the claims regarding the methodology have been significantly overstated.

Response: It is indeed true that magnetostrictive mechanisms in magnetoelectric materials have been documented, especially in those inorganic oxides. However, this magnetostrictive effect is typical weak, especially in the paramagnetic region (Ref. 24). Thus, our work is valuable considering its large magnetostrictive at room temperature. The physical origin is the synactic effect of weak exchange and soft crystalline framework of molecular magnetic ferroelectrics.

Comment 3: The compound under investigation does not seem to be a new material. Without diving too deep into the literature, also found that significant overlap with characterization data found in reference 27 provided by the authors.

Response: The structure was indeed reported in Ref. 27, in which only the dielectric, SHG, and magnetic bistability was studied. In our work, we report the ferroelectricity and ME coupling, which go much beyond Ref. 27.

In detail, the DSC, PXRD, SHG, and $\chi_M T$ are presented in Ref. 27 and the present work, all of which are basic characterizations of a ferroelectric candidate. Our results agree with those in Ref. 27. However,

these data do not touch the core issue of present work, i.e. the room-temperature magnetoelectricity. In fact, even the ferroelectricity itself had not been confirmed in Ref. 27, without the ferroelectric hysteresis loops.

In contrast, our work (especially Fig. 3) unambiguously demonstrates that it is a molecular ferroelectric material, and own significant ME coupling at room temperature at low magnetic field. Thus, our work is very important for ME coupling studying.

Comment 4: The paper contains numerous instances of contradictory data, such as: In fig. 1c, a substantial polarization value is detected at 323 K, while DSC data contradicts this observation by indicating the compound is in the paraelectric phase at this temperature. the authors demonstrate a significant polarization value detected at 323 K. Even more confusing, the data in fig. 3f indicates that the polarization value at 323 K is inexplicably 0.

Response: Thanks for this comment. As shown in Fig. S2a, the phase transition temperature of compound **1** was 338.1 K. Thus, it is ferroelectric at 323 K according to the DSC data.

Fig. R4. Comparison of SHG measurements with different configurations of temperature sensing.

In our previous measurements of SHG curves, the temperature sensing probe was placed on the top of sample, and the heating/cooling temperature scan rate was relatively fast (15 K/min) as shown in Fig. R4c, resulting in an inconsistent phase transition temperature compared to DSC measurements shown in Fig. R4a.

Therefore, we adjusted the temperature sensing probe to the surface of sample and set the scan rate of

5 K/min, as shown in Fig. R4d, the new result is basically consistent with the result of DSC measurement, as shown in Fig. R4b. We have re-measured the pyroelectricity of compound **1** under different magnetic fields (Fig. 3f). The phase change temperature is 335 K, which is same with DSC resulting. Besides, the *P-E* curve of compound **1** were re-measured at 310 K (Fig. 1c and S3).

Comment 5: The magnetostriction results obtained by In-situ single-crystal X-ray diffraction also appear inconsistent with theoretical calculation results.

Response: Thanks for this comment. Considering the paramagnetic state can not be directly mimicked in the DFT calculation, instead the lattice constants (*a*, *b*, *c*) of ground state (G-AFM) are used to qualitatively simulate the magnetostrictive effect. Although the G-AFM state is not exactly equal to the paramagnetic state in experiment, the change of lattice constants between G-AFM and FM can also reveal that there is a moderate magnetostrictive effect at zero temperature. A quantitative agreement is impossible for current computational techniques.

Comment 6: Many data provided by the author appear Inconsistencies. For instance, in fig. 1f, the display indicates 70 V, yet the accompanying icons suggest 20 V. Supplementary Table 1 present the values along axis 'a' with clear elongation, in contrast to the author's values, which appear shortened. In summary, the dataset appears messy and inconsistent.

Response: We sincerely apologize for these mistakes and improper statements. We have carefully checked full manuscript and corrected all mistakes in revised manuscript.

REVIEWER COMMENTS

Reviewer #1 (Remarks to the Author):

In the revised manuscript, the authors have made critical improvements and also made a further comparison with theoretical (DFT) calculation, all my questions and comments have been taken care of. On this basis, I think the present version now is suitable for the publication in Nature Communications.

Reviewer #2 (Remarks to the Author):

Remarks to the Author:

In the second revision of the manuscript, authors addressed all the comments raised in the earlier version. Performing DFT calculations for ferroelectric polarization confirms the credibility of the experiments as well as elevate the impact of the paper. Furthermore, referring additional supporting refereces increases the readability of the manuscript.

I would suggest this manuscript should be accepted.

Reviewer #3 (Remarks to the Author):

Hu et al.'s paper entitled, "Molecular ferroelectric with low-magnetic-field magnetoelectricity at room Temperature: Despite the authors' diligent efforts to address many of the concerns raised during the initial review process, while the subject matter of the manuscript remains intriguing, numerous unclear and even erroneous elements persist, preventing it from meeting the high standards expected by Nature Communications. For example, the compound belongs to a C₂ point group, C_m space group, I believe that's not correct. the intrinsic piezoelectricity in two direction can be seen in Supplementary fig. 4 ?, what are the two directions ? There is a distinct inconsistency between the image and the text content. The icon in Supplementary fig. 4 refers to fig. 3a, which, in reality, does not exist. The main text described the topography and reversible domain along the c-axis in Supplementary fig. 5. However, none of these actually exist in Supplementary parts. During estimating the ME coupling: fig. 3f in main text used the UCS diagram, the value is difficult to be distinguished. Supplementary fig. 11 mentioned fig. 4e, however, this also doesn't present. What is doop ? Supplementary fig. 11 reveals the peak at +1kOe is almost the same as the one at 0 Oe, this clearly contradicts the main text description.

Once again, the revised manuscript still contains many issues, and publication is not recommended.

Response to Reviewer #3

General Comment: Hu et al.'s paper entitled, "Molecular ferroelectric with low-magnetic-field magnetoelectricity at room Temperature: Despite the authors' diligent efforts to address many of the concerns raised during the initial review process, while the subject matter of the manuscript remains intriguing, numerous unclear and even erroneous elements persist, preventing it from meeting the high standards expected by Nature Communications.

Response: We thank the reviewer for his/her reading and comments/suggestions. We hope the revised version can change his/her opinion.

Comment 1: The compound belongs to a C_2 point group, C_m space group, I believe that's not correct.

Response: Thanks for this comment. We have corrected the point group from C_2 to m , while the space group is correct.

Comment 2: The intrinsic piezoelectricity in two direction can be seen in Supplementary fig. 4?, what are the two directions ? There is a distinct inconsistency between the image and the text content. The icon in Supplementary fig. 4 refers to fig. 3a, which, in reality, does not exist.

Response: Many thanks for this suggestion. We have added the intrinsic piezoelectric data along two directions (the a - and c -axis) in revised Supporting Information. The reference to fig. S3a has been updated to fig. S4a.

Comment 3: The main text described the topography and reversible domain along the c -axis in Supplementary fig. 5. However, none of these actually exist in Supplementary parts.

Response: Many thanks for this suggestion. The original Supplementary fig. 5 is the vertical PFM amplitude. We deleted Supplementary fig. 5b in revised Supporting Information. We have added the topography and domain in revised Supporting Information as Supplementary fig. 6.

Comment 4: During estimating the ME coupling: fig. 3f in main text used the UCS diagram, the value is difficult to be distinguished. Supplementary fig. 11 mentioned fig. 4e, however, this also doesn't present. What is doop? Supplementary fig. 11 reveals the peak at +1 kOe is almost the same as the one at 0 Oe, this clearly contradicts the main text description.

Response: Many thanks for this suggestion. For better view, we have revised fig. 3f. by using the data curves to replace the colour contourmap. We have corrected the reference to fig. 4e in Supplementary fig. 11 to fig. 3d. The d_{oop} denotes the slope of out-of-plane effective piezoelectric coefficient α_{31} as presented in fig. 3d. We have added the definition in the revised caption. In Supplementary fig. 11, the intensity of the peak does not relate to the ME coupling, while its position denotes a driving frequency of PFM resonance. Instead, the ME coupling can be reflected by the changes in the slope of driving electric field under magnetic field

(fig. 3d). As shown in fig. 3d, the slope decreases under the external magnetic field, implying that the piezoelectric coefficient of (TMCM)[FeCl₄] is suppressed by magnetic field.

Comment 5: Once again, the revised manuscript still contains many issues, and publication is not recommended.

Response: We thank the reviewer for his/her carefully reading and suggestions. We have carefully checked and polished our manuscript and supporting information. Hope the revised version can satisfy the reviewer.

Summary of main changes

1. On line 81, the sentence 'point group of C_2 ' has been corrected to 'point group of m '.
2. On line 111, we delete 'Supplementary Fig. 5'.
3. On line 117, 'The simultaneously acquired...the ferroelectric domain' has been changed to 'The PFM amplitude images...of ferroelectric domains.'.
4. On line 118, 'The ferroelectric properties...a domains after electrical poling.' has been added.
5. On line 149, ' d_{oop} is estimating the...from the slope of the curves' has been added.
6. On line 164, fig. 2d has been revised.
7. On line 206, 'doop' is corrected to ' d_{oop} '.
8. On line 215, fig. 3f has been revised.
9. On page 4 in Supporting Information, Supplementary fig. 1 has been revised.
10. On page 7 in Supporting Information, Supplementary fig. 4b,4d have been added.
11. On page 8 in Supporting Information, Supplementary fig. 5 has been revised.
12. On page 8 in Supporting Information, Supplementary fig. 6 has been added.
13. On page 8 in Supporting Information, 'in fig. 4e' is changed to 'in fig. 3d'

REVIEWER COMMENTS

Reviewer #1 (Remarks to the Author):

The authors have revised Fig. 3f in the manuscript, added some figures in supporting information and made further improvements for the presentation, which increases the readability of the manuscript. I think the manuscript is suitable for the publication in Nature Communications. Nevertheless, some minor points should be revised.

Minor points:

1. In Fig. 3f, the polarization decreases continuously to zero value with ascending temperature, indeed, the largest negative slope of which could manifest the ferroelectric (FE) transition. Thus, the first derivative of polarization that would display a local negative dip, is necessarily performed and plotted as an illustration in Fig. 3f. On the one hand, it will denote the FE transition directly and clearly. On the other hand, under different magnetic fields, if the position of local dip keeps unchanged, it manifests the second order FE transition. However, if the the position of local dip is changed, it implies the first order FE transition, which could be consistent with the observation on the order of FE transition from SHG measurements (Fig. 1(b) and Fig.S2(a)).
2. The formula of polarization calculated by Berry phase method should be given in method section.
3. The colour bar in Fig. S11 could be plotted, which would be more clear for the readers.

Reviewer #2 (Remarks to the Author):

While the authors have made commendable attempts to resolve the issues highlighted in the initial review, the manuscript entitled "Molecular ferroelectric with low-magnetic-field magnetoelectricity at room temperature:". Despite with high quality science and commendable work, there are still some instances where incoherency appears in the text and the figures and preventing to attain the high standards of Nature Communications.

[1] The scale bar is missing in Supplementary fig 3.

[2] The author should mention the meaning of Up and Down legends in Supplementary fig 7 and what does it refer to? Also, the explanation of these legends missing in the main text.

[3] Please make sure to have coherent units in table e in Fig 2 $H = 0$ Oe and $H = 1$ T, as written in the Supplementary tables.

[4] The author should take a brief break and instead of focusing on just the reviewer's comments, please make sure every figure contains information in the immediate text as well as the main text, especially the units. I will not recommend publishing this article at this point.

Response to Reviewer #1

General Comment: The authors have revised Fig. 3f in the manuscript, added some figures in supporting information and made further improvements for the presentation, which increases the readability of the manuscript. I think the manuscript is suitable for the publication in Nature Communications. Nevertheless, some minor points should be revised.

Response: We sincerely appreciate Reviewer 1 for the positive comments on our revised manuscript. We have made further revision to our manuscript. Point by point response are presented as below. we believe that the suggested revisions will significantly enhance the quality of our work.

Comment 1: In Fig. 3f, the polarization decreases continuously to zero value with ascending temperature, indeed, the largest negative slope of which could manifest the ferroelectric (FE) transition. Thus, the first derivative of polarization that would display a local negative dip, is necessarily performed and plotted as an illustration in Fig. 3f. On the one hand, it will denote the FE transition directly and clearly. On the other hand, under different magnetic fields, if the position of local dip keeps unchanged, it manifests the second order FE transition. However, if the position of local dip is changed, it implies the first order FE transition, which could be consistent with the observation on the order of FE transition from SHG measurements (Fig. 1(b) and Fig.S2(a)).

Response: Thanks for this valuable suggestion. Following this suggestion, we appended the first derivative of polarization of (TMCM)[FeCl₄] in the revised Supplementary Fig. 14 (also shown below). It indeed exhibits a dominant negative dip. Apparently, the position of the local dip shifts to lower temperature with increasing magnetic fields. Therefore, as pointed out by the reviewer, (TMCM)[FeCl₄] undergoes a first order phase transition, which is in good agreement with the DSC analysis.

The corresponding changes in the revised manuscript:

“In addition, the first derivative of polarization of (TMCM)[FeCl₄] exhibits a dominant negative dip (Supplementary Fig. S14). The position of the local dip shifts to lower temperature with increasing magnetic fields, which suggests that (TMCM)[FeCl₄] undergoes a first order phase transition. This finding is in good agreement with the DSC analysis.”

Comment 2: The formula of polarization calculated by Berry phase method should be given in method section.

Response: Thanks for this comment. We have added the formula of polarization in the METHODS

Part and presented as below:

where the first part \mathbf{P}_e denotes the valence electronic contribution calculated by the Berry phase method. $u_{n\mathbf{k}}$ is the wave function, the integrated interval is done over the Brillouin zone, and the sum n runs over all occupied bands. The second term \mathbf{P}_{ion} denotes the ionic contribution. e : elementary charge; Ω : volume of unit cell; Z^{ion} : charge of ion, and \mathbf{r} : position.

Note this method has been implemented in VASP and many other first-principle codes as the standard method to calculate polarization.

Comment 3: The colour bar in Fig. S11 could be plotted, which would be more clear for the readers.

Response: Many thanks for this suggestion. The colour bar is now included in the revised Fig. S11, as presented below:

Response to Reviewer #2

General Comment: While the authors have made commendable attempts to resolve the issues highlighted in the initial review, the manuscript entitled “Molecular ferroelectric with low-magnetic-field magnetoelectricity at room temperature:”. Despite with high quality science and commendable work, there

are still some instances where incoherency appears in the text and the figures and preventing to attain the high standards of Nature Communications.

Response: We thank the reviewer for his/her comments/suggestions, which greatly enhanced our manuscript. We have made further revisions in accordance with these comments.

Comment 1: The scale bar is missing in Supplementary fig 3.

Response: Thank you for this comment. However, the *P-E* and *J-E* loop curves in Supplementary Fig. 3 do not require a scale bar. Concerning the scale bar issue, we found that some PFM images were indeed missing scale bars. We have now added scale bars accordingly.

Comment 2: The author should mention the meaning of Up and Down legends in Supplementary fig 7 and what does it refer to? Also, the explanation of these legends missing in the main text.

Response: Many thanks for this suggestion. The Up and Down represent for heating and cooling measurement, respectively. We have changed the illustration in Supplementary fig 7, shown as below:

Supplementary Fig. 7 | Temperature derivative of the magnetic susceptibility of (TMCM)[FeCl₄] confirming the phase transition at 323 K. Inside: heating represents for heating measurement and cooling represents for cooling measurement.

Comment 3: Please make sure to have coherent units in table e in Fig 2 $H = 0$ Oe and $H = 1$ T, as written in the Supplementary tables.

Response: Many thanks for this suggestion. The magnetic unit is unified as “kOe”. Units in other figures are also unified.

Comment 4: The author should take a brief break and instead of focusing on just the reviewer’s comments, please make sure every figure contains information in the immediate text as well as the main text, especially the units. I will not recommend publishing this article at this point.

Response: Many thanks for this suggestion. We have carefully polished the entire manuscript to make sure that all the figures are properly referenced and described in the main text. Besides, all the identified errors have been corrected.

REVIEWERS' COMMENTS

Reviewer #2 (Remarks to the Author):

After careful review, the manuscript “Molecular ferroelectric with low-magnetic-field magnetoelectricity at room temperature” by Prof. Dong and colleagues, has shown significant improvement both scientifically and in terms of readability. I am now approving the manuscript for final publication, with just a minor observation that I have noted.

[1] On lines 224-225, the definition of $\Delta P/P_0$ is given as $(P_0-PT)/P_0$. However, the term PT does not appear elsewhere in the text. I believe the correct term should be PH, as used in the rest of the manuscript. The authors should revise this.

Response to Reviewer #2

General Comment: After careful review, the manuscript “Molecular ferroelectric with low-magnetic-field magnetoelectricity at room temperature” by Prof. Dong and colleagues, has shown significant improvement both scientifically and in terms of readability. I am now approving the manuscript for final publication, with just a minor observation that I have noted.

Response: We sincerely appreciate Reviewer 2 for the positive comments on our revised manuscript. Point by point response are presented as below. we believe that the suggested revisions will significantly enhance the quality of our work.

Comment 1: On lines 224-225, the definition of $\Delta P/P_0$ is given as $(P_0 - P_T)/P_0$. However, the term P_T does not appear elsewhere in the text. I believe the correct term should be P_H , as used in the rest of the manuscript. The authors should revise this.

Response: Thank you for this comment. We have corrected it in revised manuscript.